# Single-Molecule Localization Microscopy to Study Protein Organization in the Filamentous Fungus *Trichoderma atroviride*

**DOI:** 10.3390/molecules25143199

**Published:** 2020-07-13

**Authors:** Alexander W.A.F. Reismann, Lea Atanasova, Susanne Zeilinger, Gerhard J. Schütz

**Affiliations:** 1Institute of Applied Physics, TU Wien, Getreidemarkt 9, A-1060 Vienna, Austria; reismann@iap.tuwien.ac.at (A.W.A.F.R.); lea.atanasova@boku.ac.at (L.A.); 2Department of Microbiology, University of Innsbruck, Technikerstraße 25, A-6020 Innsbruck, Austria; 3Department of Food Science and Technology, University of Natural Resources and Life (BOKU), Muthgasse 18, A-1190 Vienna, Austria; 4Institute of Chemical, Environmental and Bioscience Engineering, TU Wien, Gumpendorferstrasse 1a, A-1060 Vienna, Austria

**Keywords:** SMLM, dSTORM, filamentous fungi, *Trichoderma*, superresolution microscopy

## Abstract

Single-molecule localization microscopy has boosted our understanding of biological samples by offering access to subdiffraction resolution using fluorescence microscopy methods. While in standard mammalian cells this approach has found wide-spread use, its application to filamentous fungi has been scarce. This is mainly due to experimental challenges that lead to high amounts of background signal because of ample autofluorescence. Here, we report the optimization of labeling, imaging and data analysis protocols to yield the first single-molecule localization microscopy images of the filamentous fungus *Trichoderma atroviride*. As an example, we show the spatial distribution of the Sur7 tetraspanin-family protein Sfp2 required for hyphal growth and cell wall stability in this mycoparasitic fungus.

## 1. Introduction

In recent years, more and more problems have arisen due to the excessive use of chemical fungicides for plant pest control. Increasing awareness is gradually changing the way agriculture is practiced, and the call for sustainable alternatives is growing. One possible solution makes use of the mycoparasitic behavior of certain fungi to protect plants from phytopathogen attack. Currently, mycoparasitic *Trichoderma* spp.—filamentous fungi that promote both plant growth and resistance as well as directly attack other plant-infesting fungi—are among the most successful biological plant protection agents [1]. Although fungal mycoparasitic behavior has been known for decades, we are only beginning to understand the underlying molecular interactions and spatial protein organizations in mycoparasitic fungi.

In the last few years, superresolution optical microscopy has enabled the analysis of the spatial distribution of biomolecules in cells at length scales of a few tens of nanometers [2]. As an example, single-molecule localization microscopy (SMLM) utilizes high precision to localize single-molecule signals in order to construct a localization map of the dye molecules bound to the biomolecule of interest. Interestingly, the literature on SMLM applied to study protein organization in filamentous fungi is scarce [3,4], likely due to experimental challenges imposed by the fungus compared to animal cells. One central problem is the strong autofluorescence signal from the fungus and from the agar used for cultivation. Commonly, researchers would revert to total internal reflection microscopy to reduce this background signal. This approach, however, is not feasible here due to the additional thickness of the cell wall, which shifts the plasma membrane of the fungus more than a hundred nanometers away from the glass surface.

To facilitate the application of SMLM to high-resolution microscopy on filamentous fungi, we screened and evaluated various imaging conditions. As a model, we used *Trichoderma atroviride* stably expressing an Sfp2-mEGFP fusion construct labeled with a dye-conjugated anti-GFP nanobody. Sfp2 has previously been identified as a tetraspanin membrane Sur7 family protein, required for hyphal growth and cell wall stability in this mycoparasitic fungus [5]. We identified three major challenges, namely, the cultivation of the filamentous fungus, blocking of non-specific binding sites, and fluorescence labeling with a photoswitchable fluorophore. In the following, we discuss the optimization of these challenges individually.

## 2. Results and Discussion

We based our study on a sample preparation protocol from Kaplan and Ewers [6] developed for the preparation of the yeast *Saccharomyces cerevisiae* for SMLM. They developed a protocol utilizing nanobodies readily penetrating through the cell wall, and thus eliminating the requirement for its prior degradation, allowing for complete structural preservation. We adapted their procedure for the filamentous fungus *T. atroviride*. Our experiments were performed on a strain constitutively expressing Sfp2-GFP. To allow for SMLM, we stained the cells with a red fluorescent AF647-conjugated anti-GFP nanobody recognizing the expressed GFP fusion construct (Figure 1A). This approach has the advantage of allowing for correlation analysis between the green GFP signal as ground truth and the SMLM data acquired from the red channel. 

A common method for monitoring filamentous fungi by inverted microscopy is to place a mycelia-covered agar block upside down onto the surface of the glass slide [7,8]; in this case, the fungus is recorded underneath the agar block. For SMLM measurements, however, the additional background signal arising from the agar block was too large to allow for high-precision single-molecule identification and localization. Using the inverted agar method, we analyzed the fluorescence background and measured a standard deviation of ~400 counts, which is approximately twice as bright as single-molecule signals recorded under the same illumination and detection conditions. Variations of the media composition were not sufficient to reduce the fluorescence background to a level that enabled SMLM. We then applied liquid media instead of agar and cultivated the fungi directly on the coverslip. Ishitsuka et al. [4] tried a similar approach to reveal a dynamic picture of cell polarity maintenance during the directional growth of *Aspergillus nidulans* using SMLM. However, in *T. atroviride* no firm hyphal attachment to the coverslip could be achieved using conidia suspensions as an inoculum, likely due to a different outer cell wall composition and more areal growth of the hyphae. We hence modified the cultivation strategy by going back to mycelia-covered agar blocks, but this time we cultivated the hyphae growing out of the agar block on the glass coverslip and measured only at positions with no agar above the hyphae (Figure 1B). For enhanced attachment of the cells to the glass bottom of the chamber, Kaplan and Ewers [6] used lectin concanavalin A (Con A) coating, which did not yield reliable results for mature hyphae of *T. atroviride*. This carbohydrate-binding protein, which is a lectin extracted from jack-beans, interacts with sugar residues (mannosyl or glycosyl) on the cell wall proteins of *S. cerevisiae* [6,9], but since the composition of cell wall glycans can differ substantially within and between fungal species, different immobilization protocols should be applied. Therefore, we additionally tested fibronectin and poly-D-lysine-coated coverslips as commonly used for cell attachment in mammalian cell lines. Especially the washing steps during the staining process force high levels of strain on the hyphal attachment, and only poly-D-lysine-coated coverslips yielded reliable attachment after the growth period or during further sample preparation. We hence used poly-D-lysine-coated coverslips for further experiments. To facilitate the growth of the fungus out of the agar block, we incubated the slides for one week in potato dextrose broth (PDB) and then used the hyphae growing out of the agar block onto the glass coverslip for fixation, staining, and imaging (Figure 1B).

For SMLM experiments, we fixed the fungal hyphae with 4% para-formaldehyde. For staining, we utilized an AF647-conjugated anti-GFP nanobody. In case of long measurement times spanning several hours, we saw a decrease of fluorescence intensity from sample to sample. We speculated that the binding of the nanobody to the GFP was not strong enough to last through the measurement procedure. Therefore, we added a second fixation step immediately after staining, which covalently bound the nanobody to the GFP. This yielded stable fluorescence intensities independent of the timespan before the measurement. To allow the nanobody to cross the plasma membrane and bind to the GFP, we permeabilized the membrane with Triton X-100 as described by Kaplan and Ewers [6]. To ensure specificity upon binding, we blocked non-specific binding by incubating with different blocking agents. As SMLM images of fungi generally show substantial background fluorescence, we took advantage of a recently developed single-molecule analysis framework, which selects specific signals based on their blinking properties [10]. 

It is of critical importance to optimize the blocking protocols for each antigen [11,12,13]. Kaplan and Ewers [6] suggested the signal enhancer Image-iT FX (Thermo Fisher Scientific, Germany), bovine serum albumin (BSA), horse serum (HS) or goat serum (GS) which were successfully applied for *S. cerevisiae*. Based on information from the manufacturer, the Image-iT FX signal enhancer largely eliminates fluorescence background due to non-specific binding, commonly seen with the application of fluorescent conjugates of streptavidin, goat anti-mouse, or goat anti-rabbit IgG. However, under the tested conditions, we could not observe a reduction of non-specific fluorescence for *T. atroviride* due to an extreme increase of autofluorescence intensity independent of the labeling. We further evaluated the remaining suggested blocking solutions and Hank’s balanced salt solution (HBSS), in addition to a mixture of equal parts of HS, GS, and BSA (Figure 2). For each composition, we compared the GFP signals with the AF647 signals by calculating the correlation coefficient between the brightness of each pixel in the GFP channel and the number of detected single-molecule signals in the AF647 channel (note that for bulk analysis in the GFP channel, cellular autofluorescence was less critical than for single-molecule analysis). Ideally, the staining protocol reaches high correlation for the specific signal and low correlation for the non-specific signal. To assess non-specific binding, we used, as a negative control, an untransformed *T. atroviride* strain not expressing the Sfp2-GFP construct (wild-type), which should not show correlated signals between the two color channels. Interestingly, we observed higher non-specific binding compared to HBSS when using the blocking reagents BSA, GS or HS, which are commonly used in SMLM protocols for mammalian cell lines. The best result was achieved by the combination of HS, GS and BSA (Figure 2B), which was chosen as the blocking agent for further SMLM measurements.

For further optimization, we next investigated different staining concentrations of the AF647-conjugated nanobody (Figure 3). For this, we calculated the number of detected localizations per µm^2^ in SMLM experiments. As a control for non-specific binding, we again used wild-type *T. atroviride*. While the non-specific signals remained very low at around five localizations per µm^2^, we found a strong increase to ~175 localizations per µm^2^ for the specific signal at a staining concentration of 25 µM. This concentration is approximately three-fold higher than the previously suggested staining concentrations for yeast samples [6,14], and orders of magnitude higher than the sub-nanomolar dissociation constant between GFP and anti-GFP nanobodies [15], indicating the presence of hindrances for the nanobody to reach its target in case of *T. atroviride*. 

Figure 4 shows three exemplary images of single mature hyphae expressing Sfp2-GFP, both in the diffraction-limited GFP channel (left) and as a localization map from the related SMLM image (right). The diffraction-limited images indicate rather homogenous staining, except for septa, which are devoid of any signal (top row). Notably, the SMLM images show clearly visible structures parallel to the hyphal axis, with a width of approximately 500 nm. We interpret these structures as multiple Sfp2-positive vesicles arrested along cytoskeletal filaments, thereby outlining their contours. In diffraction-limited epifluorescence microscopy, signals corresponding to different filaments would overlap in 2D and 3D, leading to the observed homogenous staining. In SMLM, however, there is enhanced 2D resolution. Particularly, for similar background conditions, we previously quantified the 2D single-molecule localization accuracy, yielding approximately 30 nm [10]. The comparably large structures visible in the top row of Figure 4 hence likely reflect the true extension of filament bundles of a few hundred nanometers. In addition, signals from out-of-focus planes are efficiently rejected, leading to an apparent optical sectioning effect that further improves the identification of the filaments. These findings are in accordance with membrane proteins being transported in vesicles along microtubules and actin filaments to the hyphal tip in order to sustain polarized growth [16]. Furthermore, they support our previous findings from live-cell imaging analyses revealing that Sfp2 is associated with intracellular membrane clusters and vesicles [5]. The image shown in the last row of Figure 4 is a good example of a hypha with lower Sfp2-GFP expression, yielding both a lower signal in the GFP channel (left) and fewer localizations in the SMLM image (right). 

## 3. Conclusions

The aim of this work was to develop appropriate cultivation and imaging conditions suitable for the SMLM of the filamentous fungus *T. atroviride* which overcome the problem of strong autofluorescence seen when the fungus is cultivated on solid media. This problem was solved by imaging fungal hyphae growing out of an agar block on coverslips coated with poly-D-lysine. In addition, we modified a protocol previously developed for the SMLM of *S. cerevisiae* cells. This required the blocking of non-specific binding for which the blocking reagents commonly used for mammalian cells turned out to yield unsatisfactory results. We hence demonstrated that SMLM is readily applicable to *T. atroviride* when combining different blocking reagents and applying an increased nanobody concentration. Our experiments finally allowed the visualization of vesicles containing the tetraspanin protein Sfp2 arrested along cytoskeletal filaments, supporting the strength of SMLM in providing molecular information along with super-resolved images of cellular structures. The optimized protocol will pave the way for further studies of filamentous fungi, towards a better understanding of how fungal cells function at the molecular level. Moreover, it can be directly applied to different superresolution imaging modalities, such as stimulated emission depletion (STED) microscopy or structured illumination microscopy [17]. Finally, more advanced illumination modalities such as light sheet microscopy [18,19,20] could be used to further reduce out-of-focus background signal.

## 4. Materials and Methods

### 4.1. Fungal Strains and Cultivation Conditions

Wild-type *T. atroviride* (ATCC 74058) and the Sfp2-GFP-expressing mutant [5] derived thereof were used throughout the study. All strains were maintained on potato dextrose agar (PDA) at 25 °C. 

### 4.2. Coating

For the imaging experiments, we used ibidi 2-well chambers with a glass bottom (ibidi GmbH, Martinsried, Germany). To enable the filamentous fungi to attach to the glass bottom, we coated the slides with poly-D-lysine (Sigma-Aldrich, Munich, Germany). We added 1 mL of the sterile 0.1 mg/mL solution to the glass surface and incubated them for 30 min at room temperature (RT). Afterward, the solution was removed by pipetting and the wells were dried in a sterile environment. 

### 4.3. Modified Inverted Agar Block Method

A plug of a seven-day-old wild-type *T. atroviride* or Sfp2-mEGFP-expressing mutant culture was put into the edge of each chamber (Figure 1) in a sterile environment. Wild-type and mutant strains were never cultivated on the same slide in order to avoid cross-contamination. One hundred microliters of potato dextrose broth (PDB; Sigma-Aldrich, Germany) were added, and the slides were incubated in empty sterile Petri dishes to avoid drying out at 25 °C for five days in complete darkness.

### 4.4. Fixation

For fixation, we used a 4% formaldehyde solution in a Hank’s balanced salt solution (HBSS, Sigma-Aldrich, Germany) buffer and incubated with 1 mL for 15 min. The incubation took place at RT in darkness. Two washing steps using 1 mL of 50 mM ammonium chloride in HBSS were applied for 5 and 10 min, respectively.

### 4.5. Blocking

The blocking solution was freshly prepared during the 15 min fixation process. We either used BSA, horse serum or goat serum separately with a concentration of 5 g per 100 mL HBSS and supplemented with 0.16% Triton X-100 or created a mixture of equal parts of those three solutions. HBSS as a blocking reagent was only supplemented with Triton X-100. The solution was mixed thoroughly using a vortex mixer. The sera were intended to prevent non-specific binding of the nanobodies, while Triton X-100 functions as a solvent to ensure penetration of the cell wall and the plasma membrane. We applied 1 mL of blocking solution in each chamber and incubated the slides for 30 min at RT in darkness. The same type of 1 mL blocking solution was used in the next step as a carrier solution for staining. 

### 4.6. Staining

One milliliter of the corresponding fresh blocking solution was supplemented by 25 µM of AF647 labeled nanobodies (GFP-Trap, ChromoTek, Martinsried, Germany) and was applied onto the samples. The incubation was carried out at RT in darkness for 2 h. Three washing steps with HBSS followed, each incubating for 5 min. The fixation steps were then repeated once again as described above.

### 4.7. SMLM

All microscopy experiments were carried out with a custom-made superresolution setup based on an inverted Zeiss Axiovert 200 body equipped with a Zeiss Apochromat 100×/1.45 NA oil-immersion objective (Zeiss, Oberkochen, Germany). We used a 637 nm Coherent OBIS laser for excitation and an iXon Ultra 897 EM-CCD camera (Andor, Belfast, UK). Fluorescence microscopy measurements were carried out with a freshly prepared dSTORM buffer according to van de Linde et al. [21]: 500 µL 20% glucose, 200 µL 10× PBS, 230 µL distilled water, 50 µL 1 M cysteamine, 10 µL 50 mg/mL glucose oxidase, 10 µL 12.6 mg/mL catalase. The buffer was added to each chamber immediately before imaging. For each SMLM image, we recorded 40,000 frames in epi-configuration, whereby the first 10,000 were not included in the analysis. The illumination time was 1 ms and the delay between consecutive frames was 7 ms. The raw data was pre-processed with temporal filtering to separate the slowly bleaching background from the rapidly flashing signals [10], and single-molecule signals were localized using Daostorm [22]. Finally, images were displayed utilizing a Gaussian kernel density estimator. 

## Figures and Tables

**Figure 1 molecules-25-03199-f001:**
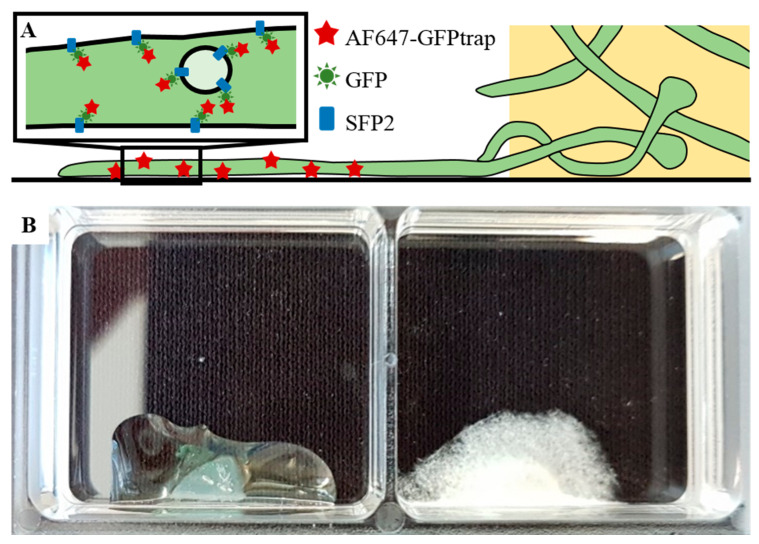
Sample arrangement for single-molecule localization microscopy (SMLM) measurements. (**A**) Schematic representation of *T. atroviride* (green) hyphae growing out of the agar block. For SMLM, GFP-fused Sfp2 protein was stained with an AF647-conjugated anti-GFP nanobody. (**B**) A sample holder showing an agar block with the medium before cultivation (**left**) and the mycelium after five days of incubation in darkness at 25 °C (**right**). This sample was then used in an SMLM experiment after fixation, blocking and staining the hyphae. The glass bottom of the 2-well chambers was coated with poly-D-lysine.

**Figure 2 molecules-25-03199-f002:**
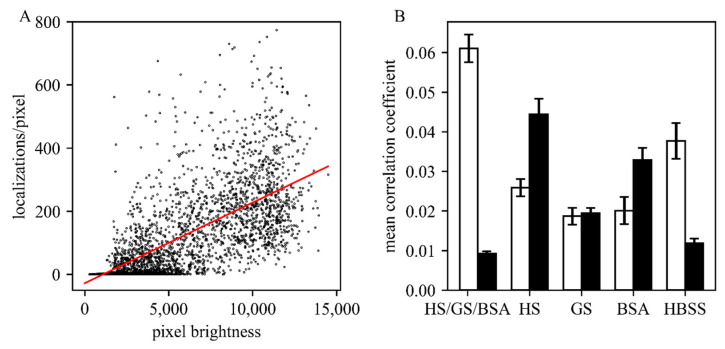
Evaluation of different blocking solutions. (**A**) Exemplary correlation analysis for 369,261 localizations obtained per pixel via SMLM are plotted versus the obtained GFP signal per pixel. In this case, a correlation coefficient of 0.026 was obtained. (**B**) Comparison of the obtained correlation coefficients for the tested (combination of) blocking solutions horse serum (HS), goat serum (GS), bovine serum albumin (BSA), and pure Hank’s balanced salt solution (HBSS) buffer (five images were analyzed for each condition). Data obtained on the Sfp2-GFP strain and wild type are shown as white and black bars, respectively. In all experiments, the concentration of AF647 was 25 µM.

**Figure 3 molecules-25-03199-f003:**
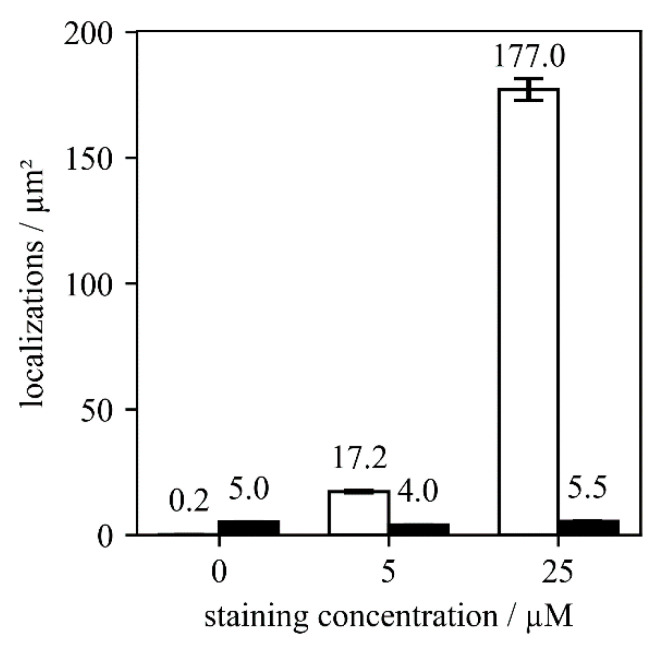
Comparison of localizations found per µm² for three different staining concentrations. We used 0, 5 and 25 µM of AF647 labeled nanobodies for wild-type *T. atroviride* (black bars) and the Sfp2-GFP strain (white bars). Five images were analyzed for each condition.

**Figure 4 molecules-25-03199-f004:**
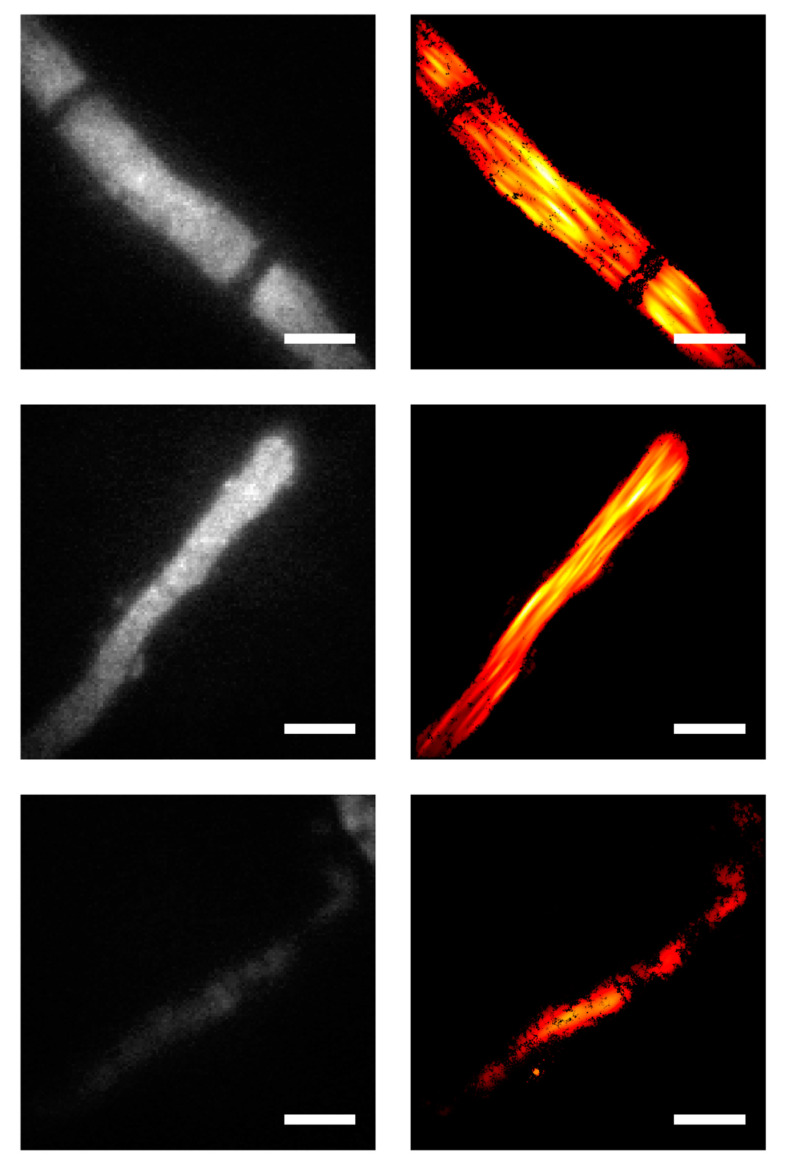
Comparison of diffraction-limited microscopy and SMLM images on *T. atroviride* hyphae expressing Sfp2-GFP. Conventional diffraction-limited fluorescence microscopy images were obtained in the GFP channel (**left column**) and compared with data recorded via SMLM at the same sample positions (**right column**). The three rows show exemplary images of single mature hyphae. Nanobody-staining concentration of 25 µM and blocking solution HS/GS/BSA were applied. Scale bar annotates 4 µm.

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
