# Peer review of "Single-Molecule Localization Microscopy to Study Protein Organization in the Filamentous Fungus Trichoderma atroviride"

_molecules, 2020, doi:10.3390/molecules25143199_

Round 1
Reviewer 1 Report
The current manuscript by Reismann et al., titled “Single Molecule Localization Microscopy to study protein organization in the filamentous fungus Trichoderma atroviride” describes the development of methods for fluorescence labeling and imaging of filamentous fungus using single-molecule localization microscopy (SMLM) using dSTORM. Super-resolution microscopy of filamentous fungi, especially T. atroviridae has been difficult, owing to the high background autofluorescence and challenges in cultivating the fungus on coverslips for imaging. Here, the authors tested various combinations of growth media, different methods for coating coverslips, cultivation of the fungus, fixation, and minimization of non-specific binding of the nanobodies to reduce the autofluorescence. By imaging the hyphal protein Spf2 that is fused to mEGFP and also an anti-GFP nanobody labeled with a red fluorophore AF647, they compared the improved resolution of using STORM to image AF647, to the diffraction-limited feature-less imaged obtained by imaging GFP. Although the resolution obtained by their imaging method does not seem very high, the method developed here is certainly a good starting point for super-resolution imaging of filamentous fungi. The manuscript is concise and well written. My comments and suggestions for further improving the manuscript are outlined below:
Comments/Suggestions:
- As the authors may be aware, background cellular autofluorescence tends to be much higher when imaging under green excitation, as compared to imaging at higher wavelengths such as using AF647. Therefore, comparison of mGFP diffraction-limited images with AF647 (as in Fig. 4) is strictly speaking, not right. Ideally, they should have used an mCherry or a far-red fluorescent protein for a more accurate comparison of resolution.
- What is the approximate resolution obtained here, as shown in Fig. 4? It does not look better than 400 nm, as judged from Fig. 4. I suggest commenting on this in the conclusions or when describing Fig. 4.
- Describe the three panels in Fig. 4 in more detail – what is being imaged in the three different rows. Why does the bottom row look very different than the top two?
- Conclusions, although are nice, I can think the authors can expand on it to comment on any other methods that can be used in the future for imaging fungi. For example, although briefly mentioned in the introduction that TIRF imaging cannot be used here, the readers can benefit from discussion on using HILO microscopy or other methods such as STED microscopy for imaging filamentous fungi.
Minor changes:
- Please italicize the atroviride species name throughout the manuscript – in the title, and in figure legends (Fig. 1 and Fig. 4)
- Single-molecule is commonly used with a ‘-‘
- Line 6: ‘a’ is missing in Austria
- Consider using ‘non-specific’ which is more accurate and widely used, instead of unspecific throughout the text.
Author Response
Comments/Suggestions:
Question: As the authors may be aware, background cellular autofluorescence tends to be much higher when imaging under green excitation, as compared to imaging at higher wavelengths such as using AF647. Therefore, comparison of mGFP diffraction-limited images with AF647 (as in Fig. 4) is strictly speaking, not right. Ideally, they should have used an mCherry or a far-red fluorescent protein for a more accurate comparison of resolution.
Answer: In principle, this is true. Yet, the contribution of autofluorescence to bulk GFP fluorescence is not as critical as in case of single molecule detection, as the bulk GFP signal levels are substantially higher than single molecule signals. In addition, spectral overlap between GFP emission and AF647 emission is smaller than for mCherry/AF647, facilitating comparative analysis. We added a clarifying statement in line 130 and in line 179.
Question: What is the approximate resolution obtained here, as shown in Fig. 4? It does not look better than 400 nm, as judged from Fig. 4. I suggest commenting on this in the conclusions or when describing Fig. 4.
Answer: We previously quantified the 2D single molecule localization accuracy for similar background conditions, yielding approximately 30 nm (see Reismann et al., Molecules 23:3338). We added a statement in line 170ff of the revised manuscript, and the reference to our previous publication.
Question: Describe the three panels in Fig. 4 in more detail – what is being imaged in the three different rows. Why does the bottom row look very different than the top two?
Answer: The three rows show exemplary images of single mature hyphae, both in the diffraction-limited GFP-channel (left) and as localization map from the according SMLM image (right). We added more description, both in the main text (line 161ff) and in the legend to Fig. 4.
Question: Conclusions, although are nice, I can think the authors can expand on it to comment on any other methods that can be used in the future for imaging fungi. For example, although briefly mentioned in the introduction that TIRF imaging cannot be used here, the readers can benefit from discussion on using HILO microscopy or other methods such as STED microscopy for imaging filamentous fungi.
Answer: We added a discussion on the implementation of the new method in light sheet microscopy and in other superresolution techniques in the conclusion section (line 202ff).
Minor changes:
Question: Please italicize the atroviride species name throughout the manuscript – in the title, and in figure legends (Fig. 1 and Fig. 4).
Answer: done
Question: Single-molecule is commonly used with a ‘-‘
Answer: changed
Question: Line 6: ‘a’ is missing in Austria
Answer: done
Question: Consider using ‘non-specific’ which is more accurate and widely used, instead of unspecific throughout the text.
Answer: done
Reviewer 2 Report
Reismann et al. have optimized a protocol for observing the localization of specific proteins inside filamentous fungi at a resolution below the diffraction limit.  They succeeded in reducing the nonspecific binding of dye-conjugated anti-GFP nanobody by optimizing the composition of the blocking agent and observed the filamentous localization of Sfp2-mEGFP fusion. This protocol would allow for high-resolution observation of organization and assembly of various proteins inside fungi. Therefore, the manuscript seems worthy of publication without major improvements.
I have only a few questions.
- Regarding the investigation of blocking solutions, the pure HBSS solution also has some blocking effect, but what happens if you mix HBSS and HG/GS/BSA mixtures?
- Although the fungal hyphae are fixed with 4% para-formaldehyde, is it possible to perform live imaging without fixing the cell? If the authors could mention about live imaging, it would be nice.
Author Response
Question: Regarding the investigation of blocking solutions, the pure HBSS solution also has some blocking effect, but what happens if you mix HBSS and HG/GS/BSA mixtures?
Answer: HG/GS/BSA was indeed mixed with HBSS, as described in the Methods section in line 232.
Question: Although the fungal hyphae are fixed with 4% para-formaldehyde, is it possible to perform live imaging without fixing the cell? If the authors could mention about live imaging, it would be nice.
Answer: True, live cell imaging would be highly Interesting. Unfortunately, there are two limitations here: first, cells need to be permeabilized for staining intracellular proteins with the AF647-labelled nanobody. This aspect has been mentioned on line 111 of our manuscript. Second, live cell imaging in combination with superresolution is additionally challenging due to the long recording time of several minutes, thereby blurring dynamics occurring during the recording time.